# Transcormer: Transformer for Sentence Scoring with Sliding Language Modeling

**Kaitao Song**[1,][*] **Yichong Leng**[2,*]**, Xu Tan**[1]**, Yicheng Zou**[3]**, Tao Qin**[1]**, Dongsheng Li**[1]
Microsoft Research Asia[1], University of Science and Technology of China[2], Fudan University[3]
{kaitaosong, xuta, taoqin, dongsli}@microsoft.com,
lyc123go@mail.ustc.edu.cn, yczou18@fudan.edu.cn

## Abstract

Sentence scoring aims at measuring the likelihood score of a sentence and is widely used in natural language processing scenarios, like reranking, which is to select the best sentence from multiple candidates. Previous works on sentence scoring mainly adopted either causal language modeling (CLM) like GPT or masked language modeling (MLM) like BERT, which have some limitations: 1) CLM only utilizes unidirectional information for the probability estimation of a sentence without considering bidirectional context, which affects the scoring quality; 2) MLM can only estimate the probability of partial tokens at a time and thus requires multiple forward passes to estimate the probability of the whole sentence, which incurs large computation and time cost. In this paper, we propose *Transcormer* – a Transformer model with a novel *sliding language modeling* (SLM) for sentence scoring. Specifically, our SLM adopts a triple-stream self-attention mechanism to estimate the probability of all tokens in a sentence with bidirectional context and only requires a single forward pass. SLM can avoid the limitations of CLM (only unidirectional context) and MLM (multiple forward passes) and inherit their advantages, and thus achieve high effectiveness and efficiency in scoring. Experimental results on multiple tasks demonstrate that our method achieves better performance than other language models. Our code and pre-trained models will be released at: `https://github.com/microsoft/CyBERTron-LM/Transcormer`.

## 1 Introduction

Sentence scoring is to measure the log-likelihood score of a sentence via language model, so that it can be used to represent the relative likeliness of a sentence. Specifically, a good sentence should have a relatively lower log-likelihood score, which means more linguistically acceptable for the sentence [1]. Due to such nature, sentence scoring has been widely used in many natural language processing (NLP) scenarios. For instance, it can be used to rerank candidates in neural machine translation (NMT) or automatic speech recognition (ASR) tasks [2, 3] or evaluate sentences in linguistic acceptability [4]. Therefore, how to design effective language models to calculate sentence scores efficiently is very important.

Recently, neural network based language models (LM) [5, 6, 7, 8] have been considered as the most widely used technique for sentence scoring, since they can produce density estimation of the whole sentence by computing the probability of each token and summing up their values as the sentence score. Specifically, causal language modeling (CLM) [9] and masked language modeling (MLM) [10] are the most representative LMs. For a given sentence $\mathbf{x} = \{x_1, \cdots, x_{|\mathbf{x}|}\}$, where $x_i$ is the $i$-th

---

[*]The first two authors are equal contributions.

36th Conference on Neural Information Processing Systems (NeurIPS 2022).

token of $\mathbf{x}$. CLM can predict next token conditioned on unidirectional context and its objective is to optimize $\sum_{i=1}^{|\mathbf{x}|} \log P(x_i|x_{<i})$. Hence, CLM is usually used for solving natural language generation (NLG) tasks [9, 11, 12]. While for MLM, it replaces a subset of tokens $\mathbf{x}_\mathcal{S}$ in $\mathbf{x}$ as special symbol [MASK] and then predicts the masked tokens based on the corrupted sequence $\mathbf{x}_{\backslash\mathcal{S}}$. The objective of MLM is to optimize $\sum_{i=1}^{|\mathcal{S}|} \log P(x_{\mathcal{S}_i}|\mathbf{x}_{\backslash\mathcal{S}})$, so it is able to learn bidirectional context and can be used for solving natural language understanding (NLU) tasks [10, 13, 14, 15]. Since CLM and MLM can be learned in an unsupervised fashion, many works [10, 13, 14, 9, 11, 12, 16, 17, 18, 15, 19, 20] have pre-trained these LMs on large-scale corpus to extract powerful linguistic representations, and these models can be directly used out of the box to predict the probabilities of tokens. Inspired by the successes of LMs, some works [21, 22, 23, 24, 25] have tried to use pre-trained CLM or MLM to generate sentence scores on NMT or ASR reranking tasks and achieved some promising results.

However, we notice that both CLM and MLM still suffer from some deficiencies in calculating sentence scores. For instance, some works [25] utilized GPT-style model for ASR reranking within single inference, yet GPT model can only extract unidirectional information due to the limitation of CLM, without considering the whole sentence semantics, and thus affect the sentence score. To utilize bidirectional context, some works [22, 23, 24] applied BERT model for rescoring. However, the nature of MLM is to mask some tokens in the sentence for prediction, which means it requires the BERT model to forward multiple times and each forward pass only masks one token for prediction. As a result, it is time-consuming to directly adopt MLM for scoring sentence. In summary, we find that for calculating sentence scores, CLM needs one-pass inference but only uses unidirectional information and MLM is costly in computing sentence score although it can use bidirectional context. Therefore, a natural question arises: is it possible to design a language model to use bidirectional context for sentence scoring with only one inference pass.

In this paper, we introduce *Transcormer*, a Transformer model designed for sentence scoring. More specifically, our Transcormer leverages a novel language modeling scheme, named as *sliding language modeling* (SLM), that produces the probability of all tokens within single inference pass and simultaneously utilizes bidirectional context. To fulfill this target, we innovatively design a triple-stream self-attention mechanism, which consists of two content streams (a forward stream and a backward stream) and one query stream. By employing specifically-designed mask strategies on the attention matrix, our method allows each token in the query stream to leverage all token information except itself (i.e., the tokens before and after it) for estimating its probability to avoid any information leakage. To the best of our knowledge, SLM is the first language modeling tailored for sentence scoring. We pre-train our SLM on large-scale corpus, and then evaluate it on multiple datasets. Experimental results demonstrate that Transcormer outperforms the baselines by up to + 0.8/0.6 BLEU score on small/large-scale NMT tasks, ~20% relative improvements on ASR tasks.

The main contributions of this work are summarized as follows:

- We analyze the pros and cons of CLM and MLM when using them for scoring sentences, and propose Transcormer with a new sliding language modeling, which uses bidirectional context for probability estimation within a single pass.

- We introduce a novel triple-stream self-attention mechanism in SLM, which has two content streams to collect forward/backward semantics, and a query stream to estimate the probability of each token in a sentence.

- Experimental results on multiple datasets demonstrate the effectiveness and efficiency of our SLM for sentence scoring.

## 2 Background

### 2.1 Sentence Scoring

Sentence scoring has a long history in NLP applications, especially in reranking tasks (*e.g.*, reranking for machine translation [26, 27, 28, 29] or speech recognition [30]). Generally, given $n$-best candidates generated by text generation models, reranking aims at scoring each candidate to select the best answer. Early works [26, 27, 28, 29, 31, 32] mainly used statistical LM or combined it with RNN-based LM to calculate the sentence scores. Recently, end-to-end neural network based LM has became the de facto approach for scoring and has been applied in many NLP tasks [6, 5, 33].

Specifically, causal language modeling (CLM) and masked language modeling (MLM) are the most representative language modeling methods, among which GPT [9, 11, 12] and BERT [10] are the most famous examples, respectively. As aforementioned, CLM conditions on the previous states to predict next token so that it can obtain the probability of each token of the sentence in a single pass, but can only capture unidirectional information. For MLM, it is able to use bidirectional context for prediction but the masked-prediction mechanism limits MLM to producing the probability of all tokens within one forward pass (since MLM can only provide the probability of masked tokens). To calculate the sentence score, a kind of solutions [22, 23, 24] is to forward multiple times and only mask one token each time. Therefore, the MLM for sentence scoring is formulated as: $\sum_{i=1}^{|\mathbf{x}|} \log P(x_i|\mathbf{x}_{\setminus x_i})$. We can find that the cost of MLM for scoring needs $|\mathbf{x}|$ inference passes, which is too time-consuming.

Some recent works [22, 21, 34, 35] have been proposed to alleviate these issues in MLM. Wang et al. [22] and Salazar et al. [21] attempted to use stochastic estimation or distillation to avoid N-passes problem to approximately estimate the probability of each toke produced by MLM, with a sacrifice of performance. Clark et al. [34] adopted a two-cloze tower [36] with noise-contrastive estimation to provide sentence probability, and Shin et al. [35] only considered word embedding as the inputs of key and value in transformer without any interaction. Besides, some works [37, 38] adopted discriminative language modeling to approximately estimate sentence scores based on the paired data, but this paradigm must require labeled data from the downstream tasks and cannot utilize unlabled data for pre-training. More discussions about related works can be found in Appendix A.2. Therefore, how to calculate sentence scores efficiently with bidirectional context is the main challenge in MLM.

Overall, CLM only needs a single forward pass to estimate the probability of all tokens but cannot extract bidirectional context, while MLM leverages bidirectional information but needs multiple inference passes. Consequently, we raise a natural question: is it possible to design a pre-trained language model to support all token prediction in a single pass and simultaneously leverage bidirectional information? This is exactly the motivation of our method.

## 2.2 Multiple-Stream Self-Attention

The pioneer of multiple-stream self-attention is XLNet [16], which introduces two-streams self-attention to incorporate autoregressive pre-training for language understanding, which pre-trains Transformer [39] via using a content stream and a query stream. In details, for the $t$-th step, the content stream is able to capture the dependency from the tokens before the $t$-th step and itself (*i.e.*, $x_{\leq t}$), while the query stream is only allowed to view the tokens before the $t$-th step (*i.e.*, $x_{<t}$) to avoid information leakage. Besides, there are some other variants of two-stream self-attention [17, 40, 41], which are designed for solving NLU tasks. For example, MPNet [17] used two-stream self-attention to build masked and permuted pre-training. ProphetNet [40] designed multiple query streams to predict N-gram future steps for sequence-to-sequence tasks, and ERNIE-GAN [41] proposed a multi-flow generation model, which includes two query streams for span and word prediction. We observe that these works mainly used a single content stream and then used one or many query streams to predict more information. Different from these works, we introduce a triple-stream self-attention mechanism, which enables query stream to leverage two content streams for prediction, and thus enjoys the benefits of additional bidirectional context for estimating token probability.

## 3 Transcormer

To inherent the advantages of CLM and MLM for sentence scoring and avoid their limitations, we propose Transcormer – a Transformer model with a novel sliding language modeling for sentence scoring. First, we summarize that an ideal language modeling for sentence scoring should satisfy two requirements: 1) model should be able to use bidirectional context for effective probability estimation of each token; 2) model should produce the probability of all tokens in a sentence within a single inference pass for efficiency. Therefore, to fulfill these two requirements, we formulate a new language modeling, named as sliding language modeling (SLM), and describe it in Section 3.1. In SLM, we propose a Triple-Stream Self-Attention mechanism based on Transformer (please see Section 3.2 for details) to use bidirectional context for each token prediction and avoid information leakage. We also discuss the differences between SLM and other LMs in Section 3.3. Figure 1 presents the pipeline of our Transcormer for sentence scoring with SLM.

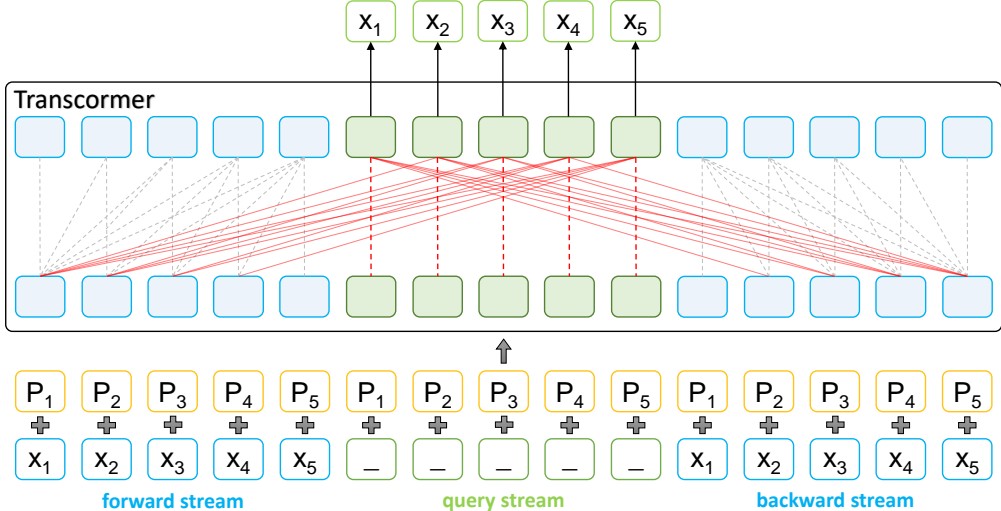

Figure 1: Transcormer with sliding language modeling. The left and the right (in blue) are forward and backward streams, respectively, and the middle (in green) is query stream. For query stream, the inputs are only the positional information. We use gray and red line to represent the allowed attended positions in the content and query streams.

### 3.1 Sliding Language Modeling

Considering the pros and cons of CLM and MLM for scoring, we notice: 1) CLM can produce probability of all tokens within one forward pass, and thus obtain the unidirectional information of the whole sentence; 2) MLM for sentence scoring needs multiple inference passes, so that many context has been repeatedly calculated and cause a waste of computation. So, is it possible to reuse token information to build bidirectional context for token prediction?

Therefore, we propose sliding language modeling (SLM) to address the inherent flaws in previous LMs (*i.e.*, CLM and MLM) for sentence scoring. Specifically, we first maintain two individual streams to collect forward (left-to-right) context and backward (right-to-left) context. And for each token prediction, we decompose the sentence information as the past tokens (the tokens before it) and future tokens (the tokens after it) respectively. As a result, our SLM enforces each token to only capture the dependency from its past tokens and its future tokens concurrently, so that each token can utilize the whole sentence information (except itself) to estimate token probability. The objective function of SLM is formulated as:

$$\mathcal{L} = \sum_{i=1}^{|\mathbf{x}|} \log P(x_i | \mathbf{x}_{<i}, \mathbf{x}_{>i}; \theta), \tag{1}$$

where $\mathbf{x}_{<i}$ and $\mathbf{x}_{>i}$ respectively correspond to the tokens before the $i$-th token and after the $i$-th token, and $\theta$ represents the parameters of SLM. Thanks to such design, our SLM can utilize bidirectional context to produce the probability of each token within one forward pass, and thus satisfy the above requirements for sentence scoring. However, previous experiences [16, 42] pointed out that the states with bidirectional information will cause information leakage when propagating to the next layer [2]. Therefore, how to implement SLM to avoid information leakage and maintain different states together is still a troublesome problem.

### 3.2 Triple-Stream Self-Attention

Based on the Eqn 1 of our proposed SLM, model needs to maintain two states to collect forward and backward contexts for prediction, and we call these two states as the forward stream and backward

---

[2]For example, assume the sequence has 3 tokens, and the hidden states of the $i$-th token at the first layer as $h_i^1$. So each position $h_1^1$, $h_2^1$ and $h_3^1$ should collect the information from positions $\{2, 3\}$, $\{1, 3\}$, $\{1, 2\}$. However, when $h_1^1$ and $h_3^1$ are delivered to $h_2^2$ at the second layer, it will cause a cyclic leakage as $h_2^2$ should not obtain information from position 2.

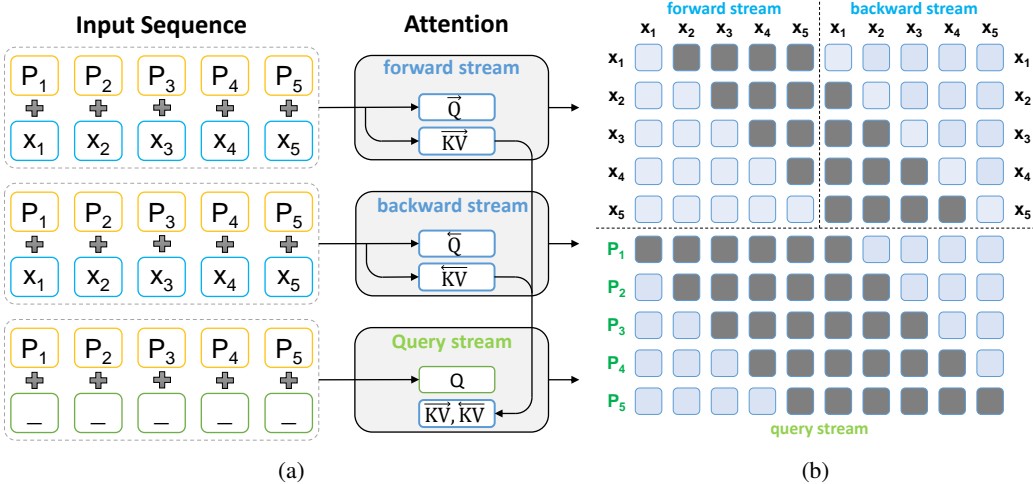

Figure 2: (a) The structure of our triple-stream self-attention used in our sliding language modeling. The query stream reuses the hidden states from both forward and backward (content) stream as the key and value in attention. (b) The attention mask matrix used in our triple-stream self-attention. The above row is the attention matrix for the forward and backward stream and the below row is the attention matrix for the query stream. The cell in gray color means this position cannot be attended.

stream respectively. To avoid information leakage, we additionally maintain an individual state for prediction and control it to only capture the dependency from the forward and backward streams. We name this state as the query stream. Therefore, we propose a novel Triple-Stream Self-Attention to update each stream, and the detailed design is described as following.

To fulfill our target, we choose Transformer [39] as our basic model, due to its flexibility in capturing global dependency. Assume the input sequence as $\{w_1, w_2, \cdots, w_n\}$ and its positions as $\{p_1, p_2, \cdots, p_n\}$, where $w_i$ and $p_i$ represents the embedding of $i$-th token and its position in the sentence, and $n$ is the token number. For the query stream, we only use the positional embeddings $\{p_1, p_2, \cdots, p_n\}$ as the input. For the forward and backward streams, we also maintain two individual states and both of them use the tokens plus its position (*i.e.*, $\{w_1 + p_1, w_2 + p_2, \cdots, w_n + p_n\}$) as the input. For the $l$-th layer calculation, we denote the forward and backward streams of the position $i$ as $\vec{h}_i^l$ and $\overleftarrow{h}_i^l$, and they are updated as:

$$\vec{h}_i^l = \text{Attention}(\mathbf{Q} = \vec{h}_i^{l-1}, \mathbf{KV} = \vec{h}_{\leq i}^{l-1}; \theta), \tag{2}$$

$$\overleftarrow{h}_i^l = \text{Attention}(\mathbf{Q} = \overleftarrow{h}_i^{l-1}, \mathbf{KV} = \overleftarrow{h}_{\geq i}^{l-1}; \theta), \tag{3}$$

where $\text{Attention}(\cdot, \cdot)$ refers to the self-attention [39] in Transformer and $\mathbf{Q}$, $\mathbf{K}$, $\mathbf{V}$ denote the query, key and value in self-attention. Hence, $\vec{h}_i^l$ collects the information before the position $i$ and itself, and $\overleftarrow{h}_i^l$ collects the information after the position $i$ and itself. For the query stream, we denote $q_i^l$ as its hidden states, and concatenate forward stream $\vec{h}_i^l$ and backward stream $\overleftarrow{h}_i^l$ of the current layer to be the key/value of query stream. So $q_i^l$ is updated as:

$$q_i^l = \text{Attention}(\mathbf{Q} = q_i^{l-1}, \mathbf{KV} = \left[\vec{h}_{<i}^{l-1}, \overleftarrow{h}_{>i}^{l-1}\right]; \theta). \tag{4}$$

Here we find that $q_i^l$ is required to only capture the dependency from the forward stream before the position $i$ and the backward stream after the position $i$, rather than itself. Due to such design, the query stream is able to capture bidirectional context for estimating token probability and avoid information leakage, which is more effective than CLM in using context. More importantly, our triple-stream self-attention enables model to predict the probability of all tokens in a sentence within a single forward pass, which demonstrates more efficiency than MLM. Figure 2 presents the detailed design of our triple-stream self-attention. We can find that in the query stream, the masked matrix is like a sliding window to control each token to view its previous states in forward stream and its future states in backward stream. And that is why we name our model as sliding language modeling.

| LM | Model | Cost | Context | Scenario |
|------|-----------|------|---------------------|----------|
| CLM | GPT [11] | $\times 1$ | forward | NLG |
| MLM | BERT [10] | $\times n$ | bidirectional | NLU |
| Bi-LM | ELMO [43] | $\times 2$ | forward + backward | NLU |
| SLM | Transcormer | $\times 3$ | bidirectional | Scoring |

Table 1: Comparisons between SLM and other LMs. We assume all LMs adopt the same architectures (*e.g.*, Transformer). The "Cost" column means the relative computations compared with CLM when calculating a sentence with $n$ tokens. The "Context" column means the contextual information usage for prediction.

### 3.3 Discussion

To better understand our SLM, we analyze the advantages of our SLM over other LMs. The comparisons are listed in Table 1. We select three representative LMs for comparisons, which are CLM (BERT), MLM (GPT) and bidirectional LM (Bi-LM, used in ELMO [43] [3]) respectively. From Table 1, we have the following observations:

1. When compared with CLM, our SLM requires $3 \times$ computations. However, our SLM can fully use the whole sentence information for prediction while CLM can only use unidirectional information. Even scaling CLM as $3 \times$ parameters, it still can not use bidirectional context for prediction. This also demonstrates the effectiveness of our SLM in using context.

2. MLM is powerful at extracting bidirectional context but it needs $n\times$ inferences to calculate the whole sentence information limited to its masked prediction. Our SLM just needs a single inference and uses bidirectional information for prediction with only $3\times$ computations. Especially, our SLM shows higher efficiency compared with MLM when $n$ is large.

3. Bi-LM can also extract forward and backward contextual information, but it just simply concatenates the forward and backward features for the final prediction, without any interactions. Instead, our SLM can iteratively fuse the bidirectional information thanks to our triple-stream self-attention mechanism.

Overall, the design of SLM is dedicated for sentence scoring, while CLM prefers NLG tasks and MLM/Bi-LM prefer NLU tasks.

## 4 Experiments

In this section, we describe our experimental setup, and the results on NMT and ASR datasets.

### 4.1 Experimental Setup

We adopt Transformer [39] as the backbone network. Following previous works [10], we adopt a base setting and a small setting for our model as Transcormer$_{base}$ with 110M parameters and Transcormer$_{small}$ with 34M parameters, that consists of 12/6 transformer layers and each layer has 768/512 hidden size and 12/8 attention heads. During the pre-training, we use wikipedia plus bookcorpus (16GB) as the training corpus, to be consistent with previous works [10]. Our model is trained at the sentence-level (*i.e.*, one sentence per sample). We choose Adam [45] as the default optimizer with learning rate of $5e-4$, $\beta_1 = 0.9$, $\beta_2 = 0.98$ and $\epsilon = 1e-6$, and weight decay is set as 0.01. The learning rate warms up over the first 10,000 steps and then linearly decays. We set the batch size as 8192 tokens per batch, and the training step is 125,000 steps. We use 32 NVIDIA Tesla 32GB GPUs, with FP16 speedup. The total training needs 5.5 days for Transcormer$_{base}$. For more experimental settings (*e.g.*, dataset and its split sizes), please refer to the Appendix. The code and pre-trained models will be released at: `https://github.com/microsoft/CyBERTron-LM/Transcormer`.

---

[3] ELMO pre-trains a left-to-right and a right-to-left LSTM [44] and concatenates the outputs of each last unidirectional LSTM layer for prediction.

| Model | IWSLT | | | | | | | | WMT |
| | De | Es | It | Nl | Pl | Ro | Ru | Tr | De-En |
|---|---|---|---|---|---|---|---|---|---|
| Oracle | 41.80 | 48.69 | 41.89 | 44.38 | 27.90 | 46.01 | 29.60 | 27.25 | 39.17 |
| Baseline | 34.77 | 41.20 | 34.95 | 37.73 | 22.67 | 38.73 | 24.21 | 21.65 | 32.54 |
| CLM (GPT) | 34.96 | 41.39 | 35.14 | 38.08 | 22.91 | 39.03 | 24.62 | 22.14 | 32.88 |
| MLM (BERT) | 35.14 | 41.54 | **35.54** | 38.14 | 23.00 | 39.21 | 24.65 | 22.36 | 33.07 |
| Bi-LM (ELMO) | 35.10 | 41.52 | 35.21 | 38.03 | 23.09 | 39.07 | 24.53 | 21.91 | 32.90 |
| SLM (Transcormer$_{base}$) | **35.24** | **41.86** | 35.52 | **38.45** | **23.29** | **39.34** | **24.69** | **22.41** | **33.10** |
| SLM (Transcormer$_{small}$) | 35.05 | 41.58 | 35.15 | 38.06 | 22.98 | 39.08 | 24.52 | 22.06 | 32.94 |

Table 2: Reranking results on IWSLT and WMT tasks, and all LMs have the same model architecture as Transcormer. The translation direction of all IWSLT tasks is to English and all results are reported in BLEU score. All LMs are pre-trained in the wikipedia + bookcorpus (16GB) with the same optimization. The last row is the oracle score from the generated candidates.

## 4.2 Experiments on Neural Machine Translation

We choose IWSLT14 dataset [46], which includes multiple small-scale translation tasks from different languages to English, and WMT14 English-German dataset [4] for evaluation. We adopt Transformer [39] as the machine translation model with 6-6 transformer layers to generate multiple candidates for reranking, with a beam size of 10. The hidden size and attention head are set as 512/1024 and 8/16 for IWSLT and WMT tasks respectively. During the reranking, we combine the original score produced by the machine translation model and the LM score with a hyper-parameter $\lambda$, by following previous experiences [7, 21]. The hyper-parameter $\lambda$ is tuned on dev set with a range of $[0.0, 2.0]$, and then select the best $\lambda$ to evaluate the test set. The results are reported in Table 2, in terms of BLEU [47]. Each task will be evaluated by five times based on different pre-trained checkpoints, and report the mean value, with a variance of 0.05. From Table 2, we have the following observations:

- Our Transcormer$_{base}$ can obtain better performance than CLM and Bi-LM [5] in both small-scale IWSLT and large-scale WMT tasks, which indicates the importance of bidirectional context for sentence scoring, and further validate the ability of our SLM in utilizing bidirectional information.

- When compared with MLM, our Transcormer$_{base}$ also achieves comparable results. Considering that the computation of MLM for scoring is linear to the input length and needs $n$ inference passes, our SLM show higher efficiency with only $3\times$ computations in a single pass to maintain three streams.

Overall, all of the results reveal that our model is more effective in using contextual information for probability estimation and more efficient with only a single forward pass, especially for long sentences.

| Model | dev-clean | dev-other | test-clean | test-other |
|---|---|---|---|---|
| Baseline | 2.80 | 6.90 | 3.06 | 7.05 |
| CLM (GPT) | 2.47 | 6.13 | 2.73 | 6.33 |
| MLM (BERT) | 2.30 | 5.65 | 2.59 | 5.90 |
| Bi-LM (ELMO) | 2.41 | 5.92 | 2.63 | 6.12 |
| SLM (Transcormer$_{base}$) | **2.23** | **5.54** | **2.49** | **5.72** |
| SLM (Transcormer$_{small}$) | 2.48 | 5.95 | 2.62 | 6.20 |
| Oracle | 1.45 | 4.23 | 1.59 | 4.19 |

Table 3: Reranking results on LibrSpeech dataset. All results are reported in WER.

---

[4]Here we only evaluate German→English direction as our model is trained on English domain.
[5]Here, we replace LSTM as Transformer in Bi-LM to keep the consistence in architecture.

| Model | Overall | ANA. AGR | ARG. STR | BINDING | CTRL. RAIS. | D-N AGR | ELLIPSIS | FILLER GAP | IRREGULAR | ISLAND | NPI | QUANTIFIERS | S-V AGR |
|---|---|---|---|---|---|---|---|---|---|---|---|---|---|
| GPT-2 (345M) | 82.6 | **99.4** | 83.4 | 77.8 | 83.0 | 96.3 | 86.3 | 81.3 | 94.9 | 71.7 | 74.7 | **74.1** | 88.3 |
| BERT (base) | 84.2 | 97.0 | 80.0 | 82.3 | 79.6 | **97.6** | 89.4 | 83.1 | 96.5 | 73.6 | 84.7 | 71.2 | **92.4** |
| BERT (large) | 84.8 | 97.2 | 80.7 | 82.0 | 82.7 | **97.6** | 86.4 | 84.3 | 92.8 | 77.0 | 83.4 | 72.8 | 91.9 |
| RoBERTa (base) | 85.4 | 97.3 | 83.5 | 77.8 | 81.9 | 97.0 | **91.4** | **90.1** | 96.2 | 80.7 | 81.0 | 69.8 | 91.9 |
| RoBERTa (large) | **86.5** | 97.8 | **84.6** | 79.1 | **84.1** | 96.8 | 90.8 | 88.9 | **96.8** | 83.4 | 85.5 | 70.2 | 91.4 |
| Transcormer (base) | 84.6 | 98.1 | 80.7 | 83.2 | 80.2 | 96.0 | 90.7 | 84.1 | 95.5 | 74.3 | 85.7 | 73.4 | 91.3 |
| + BERT Init, 20K steps | 85.0 | 98.1 | 80.0 | **84.8** | 79.2 | 96.2 | 89.3 | 84.6 | 96.1 | 76.6 | **87.8** | **75.0** | 91.7 |

Table 4: Results on BLiMP. The results on GPT-2, BERT and RoBERTa are taken from [21]. The "+ BERT Init, 20K steps" means the Transcormer model uses BERT model for initialization and then trains in SLM with 20K steps (nearly 5 epochs).

## 4.3 Experiments on Automatic Speech Recognition

We choose LibrSpeech [48] to evaluate the performance of our model for reranking on ASR task. We train a Conformer model [49] on LibrSpeech, which has 12 encoder layers and 6 decoder layers with 512 hidden size and 8 attention heads, and the beam size is set as 10. In addition, we use SpecAug [50] as a data augmentation technology to further improve the accuracy of ASR system. We use word error rate (WER) to evaluate the performance of ASR tasks. We follow the same tuning technique used in NMT tasks for hyper-parameter $\lambda$, but with a larger range as $[0.0, 5.0]$. The results are reported in Table 3. From Table 3, we find that our model can give nearly 20% relative improvements over the baseline and also outperform other LMs, including CLM, MLM and Bi-LM. The results on ASR task further demonstrates the generalization and effectiveness of our SLM in sentence scoring.

## 4.4 Experiments on Linguistic Acceptability

Following previous experiences [21], we also conduct experiments on Benchmark of Linguistic Minimal Pairs (BLiMP) [4], which includes 67K minimal pairs that contrast in grammatical acceptability and isolate specific phenomena in syntax, morphology or semantics. BLiMP provides an unsupervised setting that uses language models to evaluate sentences and the acceptable sentence can be assigned by a lower log-likelihood score. We compare our model with GPT-2, BERT and RoBERTa, and the results are reported in Table 4. We can find that our Transcormer can easily beat GPT model in BLiMP, which further validates the effectiveness of our SLM in using bidirectional context for sentence scoring. And when compared with BERT model, our Transcormer can also match or outperform BERT performance slightly. Considering that our model only needs a single pass to produce the probability of all tokens, which also manifests the efficiency of our SLM in sentence scoring. Due to the limitation of resources, our model currently cannot beat RoBERTa since RoBERTa costs more computations than BERT and thus extract better semantics for prediction. But we believe our Transcormer can achieve the better performance by training our model with the same computations.

# 5 Analyses

In this section, we conduct some method analyses on our proposed SLM and CLM/MLM. Besides, we also provide more analyses in Appendix.

## 5.1 Latency Comparisons between MLM and SLM

To better manifest the efficiency of our SLM in rescoring when compared with MLM, we further test the inference latency between SLM and MLM at both GPU and CPU under different lengths of the input sequence. Specifically, we measure the inference latency of each model at a batch size of 1 to validate the efficiency, and all results are reported in Table 5. From Table 5, we can find that MLM (BERT) produces very high latency when compared with SLM (Transcormer), especially in CPU devices. Even using a smaller BERT model, it still cannot avoid the inherent issues (*i.e.*, requires

$n\times$ computations) in MLM for scoring, which needs $20\times$ and $166\times$ additional computations over Transcormer (SLM) in GPU and CPU. These results also demonstrate the efficiency of our model in using bidirectional context for calculating sentence scores.

| Model | #Params | # Sent = 10 | | # Sent = 100 | | # Sent = 500 | |
|---|---|---|---|---|---|---|---|
| | | GPU | CPU | GPU | CPU | GPU | CPU |
| BERT (small) | 34M | 53ms | 12.87s | 502ms | 317s | 3658ms | 1,650s |
| BERT (base) | 110M | 76ms | 27.06s | 750ms | 703s | 7890ms | 3,210s |
| BERT (large) | 340M | 135ms | 59.91s | 1390ms | 1676s | 19770ms | 7,433s |
| Transcormer (small) | 34M | 59ms | 3.45s | 71ms | 7.64s | 183ms | 9.90s |
| Transcormer (base) | 110M | 67ms | 6.29s | 97ms | 17.85s | 246ms | 20.03s |

Table 5: Inference latency between BERT (MLM) and Transcormer (SLM) at different sequence lengths. "# Sent" means the length of input sequence for evaluation. GPU is evaluated at NVIDIA Tesla V100-SXM2-16GB and CPU is at Intel (R) Xeon (R) Platinum 8168 CPU @ 2.70GHz. The units of numbers in "GPU" and "CPU" columns are at millisecond (ms) and second (s).

## 5.2 Comparable Computation between CLM and SLM

As aforementioned, our SLM needs $3\times$ computations when compared with CLM. To make a fair comparison, we also pre-train a Transcormer$_{small}$ with 34M parameters in total, which consists 6 transformer layers and each layer has 512 hidden size and 8 attention heads. Hence, our Transcormer$_{small}$ has approximately $\frac{1}{3}$ parameters of Transcormer$_{base}$, and has the similar computations as GPT$_{base}$. We conduct experiments on three NMT tasks and an ASR task (LibriSpeech dataset) for comparisons and the results are listed in Table 2 and 3 (*i.e.*, "Transcormer$_{small}$ row"). We can find that even under the same computation, our model still outperforms CLM and this result further validates the necessity of using bidirectional context for sentence scoring. Besides, considering that Transcormer$_{small}$ has fewer parameters, our model is also friendly to the device deployments (*e.g.*, CPU).

| Model | Cost | PPL | dev-clean | dev-other | test-clean | test-other |
|---|---|---|---|---|---|---|
| Baseline | - | - | 2.80 | 6.90 | 3.06 | 7.05 |
| MLM ($k=1$) | $\times n$ | 4.26 | 2.30 | 5.65 | 2.59 | 5.90 |
| MLM ($k=2$) | $\times \lceil n/2 \rceil$ | 8.41 | 2.41 | 5.87 | 2.70 | 6.20 |
| MLM ($k=3$) | $\times \lceil n/3 \rceil$ | 11.58 | 2.60 | 5.95 | 2.87 | 6.41 |
| MLM ($k=\frac{n}{3}$) | $\times 3$ | - | 2.75 | 6.71 | 2.98 | 6.93 |
| MLM ($k=\frac{n}{2}$) | $\times 2$ | - | 2.80 | 6.80 | 3.01 | 6.99 |
| SLM | $\times 3$ | **3.85** | **2.23** | **5.54** | **2.49** | **5.72** |

Table 6: Comparisons of sampling different $k$ tokens for prediction on MLM. We choose ASR reranking task on LibriSpeech dataset to evaluate the results and also report PPL on a subset of sentences with same length ($n=20$).

## 5.3 Varying Numbers of Forward Passes in MLM

As mentioned above, MLM needs to forward $n$ passes as each time only mask one token. So what will happen if we allow each pass to mask more tokens? Therefore, we design experiments that enforces MLM to forward $k$ tokens for prediction so that it only needs $\lceil n/k \rceil$ passes, and investigate the effect of different $k$. For a certain $k$, we randomly split the sentence as $\lceil n/k \rceil$ sets, and each time masks one subset for prediction. The comparisons are listed in Table 6. We find that using larger $k$ will severely harm the performance, even if it can reduce the number of inference passes. When $k$ is set as $\frac{n}{3}$, which is equal to the cost of our SLM, it can hardly give any improvements over the baseline. We think that masking more tokens in the sentence will make it more difficult to estimate the token probability at a time. These comparisons also highlight the efficiency and effectiveness of our SLM for sentence scoring.

### 5.4 Sentence Scoring Quality at Each Position

Following previous experiences [21], we count the cross-entropy loss of each position in sentences for each LM, to better analyze the effectiveness of bidirectional context to estimate token probability. Specifically, we sample a subset of sentences as $\mathcal{S}$, and each sentence has the same token number $n$ (here $n$ is 20). For each position $i$, we count the average cross-entropy over the sampled subset $\mathcal{S}$ based on the output probability of each LM. The results are displayed in Figure 3. We can find that: 1) For CLM, the cross-entropy loss is higher at the first several positions and gradually decreases for the subsequent positions but is still higher than MLM and SLM, which indicates that only using undirectional information is not enough to measure the sentence score precisely. 2) SLM can almost obtain similar loss as MLM at each position. Considering SLM just needs a single pass while MLM needs $n$ passes, this phenomenon further validates the superiority and efficiency of SLM in scoring sentences.

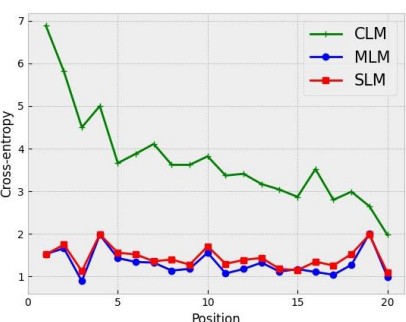

Figure 3: The average cross-entropy loss of each LM at each position. MLM uses $n$ passes to predict each position.

## 6 Conclusion

In this paper, we propose Transcormer, a Transformer with a novel sliding language modeling for sentence scoring. Specifically, our SLM is able to produce the probability of each token over the whole sentence within a single forward pass, and utilizes bidirectional context for prediction, and thus inherents the advantages of CLM and MLM and avoids their deficiencies. To the best of our knowledge, the proposed Transcormer is the first pre-trained language model tailored for sentence scoring. Experimental results on multiple datasets demonstrate the effectiveness of our Transcormer in computing sentence score for reranking tasks.

Besides, we summarize some potential directions of our Transcormer and SLM as the future works:

- Currently our Transcormer is only conducted on the English domain under the base setting, due to limited computation. We expect to develop large-scale Transcormer and use different language domains or multilingual data for training in the future.

- We design sliding language modeling for sentence scoring, and our experiments are mainly on reranking task. However, based on the characteristics of our SLM, we believe our model can also be used for other scenarios (*e.g.*, error correction [51, 52, 53], data selection), and we will explore the specific fine-tuning techniques when applying our SLM on different downstream tasks.

- Besides, our Transcormer mainly pre-trains SLM on an encoder framework. However, our SLM is not limited to model structure. For example, SLM can be easily extended to encoder-decoder framework [18] based on paired data. Therefore, we also expect to explore the possibility of using SLM on different frameworks.

- Although our paper mainly focuses on text data, we want to highlight that SLM can also be extended to other different modalities with sequential characteristic (*e.g.*, image, speech [54] and time series data). Consequently, how to apply SLM to other modalities is also a valuable topic in the future.

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
