# A  Appendix

## A.1  Experimental Setup

### A.1.1  Datasets

**IWSLT 2014** is the evaluation campaign of the 11th International Workshop on Spoken Language Translation. It consist of a lot of small-scale translation tasks collected from TED talks, including German (De), Spanish (Es), Italian (It), Dutch (NL), Polish (PL), Romanian (Ro), Russian (Ru), Turkish (Tr) to English. We randomly split each dataset as the training set and dev set with a ratio of 25:1. And each task concatenates TED.tst2010, TED.tst2011, TED.dev2010 and TED.tst2012 as the test set. The statistics of each sub-task is described as:

|       | De   | Es   | It   | Nl   | Pl   | Ro   | Ru   | Tr   |
|-------|------|------|------|------|------|------|------|------|
| Train | 160K | 169K | 167K | 153K | 128K | 167K | 153K | 109K |
| Valid | 7.2K | 7.6K | 7.5K | 6.9K | 5.8K | 7.6K | 6.9K | 4.9K |
| Test  | 5.5K | 5.5K | 5.5K | 5.3K | 5.4K | 5.5K | 5.5K | 5.4K |

Table 7: Statistical of IWSLT datasets.

**WMT14 English-German** comprises 4.5M bilingual data collected from Europarl v7, Common Crawl corpus and News Commentary. We concatenate newstest2012 and newstest2013 as the valid set, and choose newstest2014 as the test set for WMT14 English-German. Our experiments mainly focuses on German→English.

**LibriSpeech** [48] includes 1000hr speech data, sampled at 16k Hz. LibriSpeech includes four subsets for evaluation, which are dev-clean, dev-other, test-clean and test-other.

| Hyper-parameter   | Transcormer$_{base}$ | Transcormer$_{small}$ |
|-------------------|----------------------|-----------------------|
| Number of Layers  | 12                   | 6                     |
| Hidden Size       | 768                  | 512                   |
| Filter Size       | 3072                 | 2048                  |
| Attention heads   | 12                   | 8                     |
| Dropout           | 0.1                  | 0.1                   |
| Weight Decay      | 0.01                 | 0.01                  |
| Learning Rate     | 5e-4                 | 5e-4                  |
| Steps             | 125K                 | 125K                  |
| Batch             | 8192                 | 8192                  |

Table 8: Pre-training hyper-parameters for Transcormer$_{base}$ and Transcormer$_{small}$.

### A.1.2  Hyper-parameter Setup

The pre-training hyper-parameters of Transcormer are described in Table 8.

## A.2  More Details about Related Works

As mentioned in Section 2.1, some works [22, 21, 34, 35] tried to alleviate the efficient problem in MLM model caused by N-passes. Specifically, [22] proposed to calculate pseudo log-likelihood score via stochastic estimation, that is randomly sampling K tokens and computing the probability of these K tokens via masked prediction as the final sentence probability. It can reduce the time complexity from $O(|\mathbf{x}|)$ to $O(K)$ but will harm model performance. [21] also suggested a distillation strategy to cover this problem, that requires model to compute sentence score via N-passes first (*i.e.*, teacher model) and then distills it to the output vector of the [CLS] token (*i.e.*, student model) during the pre-training. However, this paradigm also will under-perform regular LMs in their experiments [21]. [35] designed a model named DLM to produce token-wise probability via a single inference pass. To fulfill this target, DLM only feeds word embeddings as the key/value for each Transformer layer, rather than the previous layer. Such design allows the query stream to capture the whole

| Model | Domain | Dev | | Test | |
|---|---|---|---|---|---|
| | | clean | other | clean | other |
| Baseline [23] | - | 7.17 | 19.79 | 7.26 | 20.37 |
| Electric [34] | wikibooks | - | - | 5.65 | 17.42 |
| T-TA [35] | Libri | 4.98 | 16.09 | 5.11 | 16.91 |
| GPT-2 (117M) | openwebtext | 5.39 | 16.81 | 5.64 | 17.60 |
| GPT-2 (345M) | openwebtext | 5.15 | 16.48 | 5.30 | 17.26 |
| BERT (base) | wikibooks | 5.17 | 16.44 | 5.41 | 17.41 |
| RoBERTa (base) | RoBERTa [13] | 5.03 | 16.16 | 5.25 | 17.18 |
| Transcormer (base) | wikibooks | 5.09 | 16.30 | 5.28 | 17.31 |
| + BERT Init, 20K steps | wikibooks | 5.10 | 16.27 | 5.19 | 17.20 |
| Transcormer (base, 20K steps) | Libri | 4.61 | 15.51 | 4.73 | 16.45 |

Table 9: WERs on LibriSpeech after rescoring. We evaluate the results on the dataset shared by [23], and the results of baseline, GPT and BERT are taken from [23]. RoBERTa adopts a large-scale pre-training corpus with 160GB.

sentence information but without any contextualized semantics. Electric [34] is a model built upon a Two-Cloze Tower [36], based on noise contrastive estimation. More in details, Electric trained a left-to-right Transformer and a right-to-left Transformer and then concatenated together at the final layer to predict each token. Just as discussed in Section 3.3, this model learns forward and backward context individually and only fuse semantics at the final layer while the query stream of our SLM is able to fuse bidirectional context iteratively. Overall, our SLM can make full use of bidirectional information over all Transformer layers and predict token-wise probabilities simultaneously.

Besides, there also remain some works [37, 38] which use discriminative language modeling to approximately estimate sentence scores. [37] borrowed the idea of the energy-based model into sentence reranking. [38] proposed a discriminative language model that minimizes the KL-divergence between the target distribution and the output distribution. These methods can be considered as the discriminative language modeling which directly predicts a single value to be the sentence score. Discriminative language model usually needs target datasets for fine-tuning, while our proposed language model is independent to downstream tasks. We think discriminative language models are complementary to our works and we leave this combination as the future works.

## A.3  Results

### A.3.1  Comparison with other works

As aforementioned, previous works [35, 34] have tried some strategies to calculate the probabilities of all tokens simultaneously to avoid the limitations in N-passes. To validate the advantages of our model in using bidirectional context, we also conduct experiments to make a comparison with these methods. For the sake of fairness, all of our experiments are deployed on the same datasets used in [23] and the results are shown in Table 9. From Table 9, we find that our Transcormer can outperform these models. These improvements also demonstrate the superiority of our model in utilizing bidirectional context to predict sentence probability.

## A.4  Accelerating SLM Training

Although SLM is more advantageous than CLM in using bidirectional context and demonstrates higher efficiency than MLM for sentence scoring, directly training SLM from scratch is still time-consuming since it needs $3\times$ computations to maintain query and forward/backward streams. So is it possible to accelerate the training of SLM? We note that both SLM and MLM adopt the masked token plus its position to predict its target based on the context while the main difference in context is that MLM adopts one bidirectional context and SLM adopts forward and backward contexts. We think that MLM-based model with bidirectional context should also be able to produce good unidirectional context and SLM does not modify model structure, so that we can use the MLM-based model (e.g.,

BERT) for initialization to accelerate SLM training. Specifically, we continue to train BERT model with addition 5 epochs (nearly 20K steps) and the results can be found in Table 4 and Table 9. We can find that only needs 5 epochs, our Transcormer with BERT model initialization can match the performance that is trained from scratch, which means using BERT initialization can accelerate our SLM training.

## A.5 Analyses

### A.5.1 Pre-training Strategy

In our experiment setup, we use sentence-level data as the input for pre-training. To better analyze the effect of using different data processing for pre-training, we conduct experiments by using stream-level data (concatenate multiple sentences as a fixed-length, *e.g.*, 512) to make a comparison. We apply two strategies on NMT & ASR tasks, and then evaluate the average (top-1) accuracy of each token in a sentences with different lengths based on the output probability for SLM. The results are reported in Table 10. We can find that using stream-level data for pre-training can not achieve good accuracy when sentence length is too short. We guess that is because using stream-level data causes model can not fit short sentence since it always pre-trains under the longer sentences (*i.e.*, 512). Considering that our downstream scenarios mainly consist of single sentence, which is usually too short, directly using stream-level data for pre-training can not achieve promising performance. As a result, we recommend to use sentence-level data for pre-training, and we also expect to explore more effective pre-training strategies in the future.

| Model | IWSLT | | WMT | LibriSpeech | | # Sent Len | | |
| | De | Es | De-En | dev-clean | dev-other | 20 | 250 | 500 |
| --- | --- | --- | --- | --- | --- | --- | --- | --- |
| Transcormer | 35.24 | 41.86 | 33.10 | 2.23 | 5.54 | 60.0% | 73.0% | 78.8% |
| Using stream-level | 34.84 | 41.38 | 32.70 | 2.56 | 6.31 | 20.0% | 55.0% | 78.5% |

Table 10: Comparisons between sentence-level and stream-level pre-training. The translation direction of all IWSLT tasks is to English. We sample some sentences from wikipedia with the same length (*e.g.*, 20, 250, 500) to evaluate the token accuracy of SLM in sentences (*i.e.*, obtain the top-1 accuracy of each token and calculate the sentence accuracy by averaging the accuracy of all tokens).

### A.5.2 Domain Adaption

Following previous experiences [21], we also study the effect of using in-domain data for pre-training. For NMT tasks, we randomly sample 20GB monolingual data from NewsCrawl data to build the pre-training corpus for pre-training. And for ASR tasks, as LibriSpeech includes 4GB in-domain data, we direct use this data as our pre-training corpus to handle ASR tasks. The results of NMT and ASR tasks are reported in Table 11 and Table 12. We can find that using in-domain data for pre-training is useful to improve the downstream tasks.

| Model | IWSLT | | | | | | | | WMT |
| | De | Es | It | Nl | Pl | Ro | Ru | Tr | De-En |
| --- | --- | --- | --- | --- | --- | --- | --- | --- | --- |
| Oracle | 41.80 | 48.69 | 41.89 | 44.38 | 27.90 | 46.01 | 29.60 | 27.25 | 39.17 |
| Baseline | 34.77 | 41.20 | 34.95 | 37.73 | 22.67 | 38.73 | 24.21 | 21.65 | 32.54 |
| SLM (Transcormer) | 35.24 | 41.86 | 35.52 | 38.45 | 23.29 | 39.34 | 24.69 | 22.41 | 33.10 |
| + in-domain data | **35.74** | **42.39** | **35.97** | **39.06** | **23.91** | **39.70** | **24.95** | **23.05** | **33.51** |

Table 11: Domain adaption on NMT tasks. The translation direction of all IWSLT tasks is to English. All results are reported in BLEU.

## A.6 Can SLM be used for Language Understanding Tasks?

As aforementioned in Section 3.3, both SLM and MLM can learn bidirectional context to predict token-wise probability, so why we cannot directly use SLM for language understanding tasks? First,

| Model | dev-clean | dev-other | test-clear | test-other |
|---|---|---|---|---|
| Oracle | 1.45 | 4.23 | 1.59 | 4.19 |
| Baseline | 2.80 | 6.90 | 3.06 | 7.05 |
| SLM (Transcormer) | 2.23 | 5.54 | 2.49 | 5.72 |
| + in-domain data | **2.01** | **5.12** | **2.12** | **5.23** |

Table 12: Domain adaption on LibrSpeech dataset. All results are reported in WER.

| **Prompt Pattern** | | **SST-2** |
|---|---|---|
| Accepted Sentence | Unaccepted Sentence | (zero-shot) |
| The sentiment of [**Review**] is [**GT Label**]. | The sentiment of [**Review**] is [**Wrong Label**]. | 58.0% |
| [**Review**] is a [**GT Label**] sentiment. | [**Review**] is a [**Wrong Label**] sentiment. | 71.0% |

Table 13: Examples of Ranking classification for solving sentiment classification. "[**Review**]" means the input sentence and "[**GT Label**]" means the real label of the input sentence and "[**Wrong Label**]" means other incorrect labels. The last column is the zero-shot results on SST.

we want to highlight that although the query stream is able to enjoy completely bidirectional context like MLM due to the design of the triple-stream self-attention mechanism, there still remain some differences between MLM and SLM. First, the query and content streams in the MLM-based model are shared together to enjoy the benefits of the bidirectional context, while our SLM maintains query and forward/backward streams individually. In other words, the content in MLM can learn bidirectional context while SLM is used to collect forward and backward context. Considering previous state-of-the-art works [10, 13, 16, 14] have proven that BERT-style models can be fine-tuned to obtain superior performance in NLU tasks due to the benefits of bidirectional context. We think that MLM prefers NLU tasks while SLM is more suitable for scoring.

In our internal experiments, we have tried to directly fine-tune SLM on SST-2 [55] like BERT (*i.e.*, using bidirectional context). Due to the mismatch (*i.e.*, forward/backward context v.s. bidirectional context) between pre-training and fine-tuning, our SLM only achieves 91.9% accuracy while the standard BERT can obtain 92.8%. This phenomenon may also validate our hypothesis that SLM is not the optimal method for solving NLU tasks in a BERT-style fine-tuning paradigm.

However, there still remain some alternative possibilities for SLM to adapt NLU scenarios. A possible solution is that converts language understanding tasks into ranking classification tasks. For example, assuming the task is sentiment classification and we have a review sentence, so we can construct two sentences: "the sentiment of [review] is positive" and "the sentiment of [review] is negative". Based on the constructed sentences, we can use our model on each sentence to calculate their sentence scores and the sentence with ground truth should have a lower log-probability score. It can be considered as a variant of prompt-based learning [12, 56]. We have simply built two different patterns to formulate sentiment classification, and conduct experiments on SST-2 in a zero-shot setting (*i.e.*, without fine-tuning). The results are shown in Table 13. We find that using patterns like "[**Review**] is a [**GT Label**] sentiment." can obtain an accuracy of 71.0% on SST-2 in a zero-shot setting (BERT fine-tuning cannot be used in a zero-shot scenario). These experiments also demonstrate that we can transform NLU tasks into ranking tasks and then apply our model for ranking, which also indicates the potential of our model in solving NLU or other NLP tasks. We will also continue to explore more techniques to refine this paradigm to improve performance in the future.