# OpenReview forum: "Transcormer: Transformer for Sentence Scoring with Sliding Language Modeling"
_NeurIPS.cc/2022/Conference — NeurIPS 2022 Accept_

### Official Review · Reviewer_ayVD · 2022-07-10

**Rating:** 6
**Confidence:** 5
**Soundness:** 4 excellent
**Presentation:** 2 fair
**Contribution:** 2 fair

**Summary:**

This work proposes a modified Transformer encoder, the Transcormer, which gives per-position log (pseudo-)probabilities in a single pass (unlike masked LMs) while still capturing token-token dependencies at all layers. These probabilities sum to give a score that can be used to rescore NMT and ASR hypotheses. This works by having the key-value representations be concatenated from a forward (inclusive) and backward (inclusive) attention stream before taking the outer product with the query, which never sees the content of its own position. Experiments on IWSLT, WMT, and LibriSpeech suggest this scheme outperforms MLM and CLM for rescoring.

**Post-rebuttal: I raise soundness to 4/4 and my score from 4/10 to 6/10.** The work is thorough and introduces an architectural innovation to more richly model $P(y_s|y_{\backslash s})$ in one pass. Unlike the paper originally claimed, it is not the first such model (T-TA and Electric); also, unsupervised sentence scoring remains niche (MLM scoring is more for inspection of pre-existing models, and for downstream tasks most works use discriminative ranking). However, the authors have done many experiments, it performs better than MLMs and previous models, and may have applications elsewhere (analogous to XLNet, also published at NeurIPS). There are still many grammar errors; please work with a copyeditor or native English speaker on any camera-ready.

**Questions:**


- Could you compare and contrast your approach with other one-pass models (T-TA, Electric)?

- Given only the sum of probabilities is ever used and since one must pretrain anyways, have you compared with the stochastic estimation or distillation methods (L96-L97)? Alternately, do you have an application where token-wise probabilities are essential?

- Will you release your pretrained SLMs?

- L199-200 (SLM for scoring, CLM for NLG, MLM for NLU) is not really explained -- are there experiments for this? esp. since MLM and SLM model the same quantity.

- Have you considered ways of matching MLM and SLM costs that avoid MLM pretraining mismatch (and how do you formally define cost), e.g., a shallower MLM, lower attention dimensions? Alternately, what if the MLM were pretrained with masked sets?


**Limitations:**

While SLM gives token-wise probabilities out of the box, it does require pretraining from scratch and is rather specialized (whereas MLM and CLM have other uses). Hopefully more uses can be explored to justify this cost; see works above for examples, like acceptability (BLiMP) or text similarity (STS).

**Strengths And Weaknesses:**

The authors succeed in proposing a scheme that models token probabilities conditioned on all other tokens (Eq. 1), in a single pass, while preserving "deep bidirectionality". The method appears sound, and while the mechanism is complicated, Figures 1 and 2 do well to explain it (though Figure 2b should have labels saying which axis is layer $l-1$ and which is layer $l$).

However, I have concerns re: the clarity and significance of this work.

Firstly, there are two uncited works that have proposed one-pass solutions to token-wise probabilities:
- T-TA: J Shin, Y Lee, S Yoon, K Jung, “Fast and Accurate Deep Bidirectional Language Representations for Unsupervised Learning”, ACL 2020
- Electric: K Clark, M-T Luong, Q Le, C Manning, "Pre-Training Transformers as Energy-Based Cloze Models", EMNLP 2020

In particular, the first work's proposed T-TA has many similarities:
- a query initialized with position embeddings only
- a query that is updated through to the end and is selectively blocked on the self position

while having some benefits (no 3x in cost?) and disadvantages (the keys, values do not transform over layers).

The authors should also acknowledge [18] in section 5.3, where an almost identical analysis is performed, even down to the choice of length and plot (Figure 3 here is largely [18]'s Figures 3+4).

While the authors compare on some tasks (NMT, ASR) as prior work, it would be good to compare on the same datasets or even n-best lists. For example, [18], [20], and the above two works compare on a shared set of n-best LibriSpeech lists while this work creates new ones. This is mitigated by the authors releasing model code (thank you!), but would help convince readers -- especially as SLM models are not off-the-shelf but trained from scratch (and hopefully released).

It is also unclear e.g., what L20's "quality of a sentence" or what L23-24's "precise sentence scoring" mean. Only in Section 2.1 (page 3) do we get two definitions:
- via CLM, which by the chain rule gives the log-likelihood score $\log P(y_1, ..., y_{|y|})$
- via MLM, which gives the pseudo-log-likelihood score $\sum_{i=1}^{|y|} \log P(y_i \mid y_{\backslash y_i})$

SLM models the latter. It would be better to make this clear early, and *then* mention how/why these quantities are empirically (or even theoretically?) good quantities to rescore with (so before in 26-38), with the second being better (which is why(?) SLM models it). Discriminative rescoring should also be discussed/contrasted early, or at least in the main text (not the appendix).

I appreciate some of the analyses which complement past work, in particular A.3.1's discussion comparing sentence vs. fixed-length pre-training, and 5.3 showing SLM and MLM's similar scores. Section 5.1 and 5.2 are unsatisfying though, as increasing the number of masked tokens would be expected to be poor given the mismatch with pretraining (in BERT, masking 15%).

Minor notes:
- The word "sliding" evokes sliding windows (i.e., only a contiguous, fixed length of inputs are considered at any time), giving the wrong impression
- L195-198: The structure of bi-LMs should be explained more
- L163: should clarify that $p_i$ refer to position _embeddings_

Writing should be proofread, e.g.,
- L18 and elsewhere: "language modelings" --> "language modeling approaches" or "language modeling schemes" or even "language models"
- L85: "in a chain-style rule" --> "via the chain rule"
- Figure 2: "reuse" --> "reuses"; "can not" --> "cannot"
- Table 5, Appendix Table 5: "test-clear" --> "test-clean"
- L233, 236, Appendix Table 5: "LibrSpeech" --> "LibriSpeech"

---

> ### Author Response · Authors · 2022-08-02
> **To Reviewer ayVD**
>
> **Q1: Two uncited works and need to be compared.**
>
> Thanks for pointing out our missing references. We have added the corresponding descriptions of these two works in the related works in the revised version. Due to the page limitations, the comparisons between these two works are put in Appendix A.3. Besides, the differences between these works are as follow:
>
> - The motivation between T-TA and SLM is similar, while the main difference is that the key/value of each transformer layer of T-TA is always from word embedding, rather than the previous layer. For SLM, the key/value in forward and backward stream can collect left-to-right and right-to-left contextualized semantics, which can abstract better representations compared with word embeddings.
>
> - Electric is built upon the Two-Cloze Tower model, with noise contrastive estimation. Two-cloze tower model is similar to Bi-LM as described in our paper. It first trains a left-to-right Transformer and a right-to-left Transformer and then concatenates together at the final layer to predict each token. Just as discussed in Section 3.3, this model just uses forward context and backward context individually and fuses forward/backward context at the last layer while our SLM can iteratively fuse bidirectional semantics.
>
> The results in Table 8 also prove the ability of our model in learning bidirectional context.
>
>
> **Q2: Acknowledge [18] in section 5.3.**
>
> Thanks for your advice. We have acknowledged [18] in section 5.3 in the revised version.
>
> **Q3: Compared on the same datasets used in [18][20].**
>
> Thanks for your advice. We have added the related comparisons with different models in Appendix A.3 on the same datasets used in [18][20] for the sake of fairness.
>
> **Q4: More clear definitions about SLM.**
>
> Thanks for your question. We have adjusted the descriptions in the introduction and added more related works in the background. Following your advice, we have:
>
> 1. The definitions of CLM and MLM have been moved to the introduction.
>
> 2. To avoid confusion in your understanding, we have modified the descriptions of “quality of a sentence” and “precise sentence scoring”. Specifically, sentence scoring is to evaluate the log-likelihood score of a sentence, and it can reflect the goodness of a sentence. That means a good sentence should be assigned with a lower log-likelihood score while a bad sentence is higher.
>
> **Q5: Discriminative Rescoring should be discussed early.**
>
> Thanks for your suggestion. We have moved the descriptions about discriminative rescoring into the main text (Line. 99 - 100). Due to the page limitations, more details about related works are put in Appendix A.2.
>
>
> **Q6: Minor notes and writing.**
>
> Thanks for your suggestions. We have revised and proofread the writing of our paper in the revision version and provided more explanations about some definitions.
>
> **Q7: Have you compared with the stochastic estimation or distillation methods**
>
> Thanks for your questions. Generally, previous works [a][b][c] have indicated that using stochastic estimation or distillation methods are two compromising strategies to avoid the expensive computation in obtaining pseudo log-likelihood scores with a sacrifice of model performance.
>
> To resolve your concerns, we also conduct experiments to compare our model with stochastic estimation. We do not compare distillation methods since it requires the model to calculate sentence scores via N-passes (i.e., we need to calculate the pre-training corpus N times) first and distill the sentence score to the output vector of CLS token during the pre-training, which needs enormous computations and cannot be accepted even it can be pre-calculated. Stochastic estimation is to randomly sample K tokens and calculate K tokens’ probability via MLM as the final sentence score, so that it can reduce the time complexity from O(n) to O(K). Here, we simply evaluate the performance of stochastic estimation to compare it with our model.
>
> | Model | cost|dev-clean | dev-other | test-clean | test-other |
> | :----- | :----:|:----: | :----: |:----: |:----: |
> | Baseline | -| 7.17 | 19.79 | 7.26 | 20.37 |
> | BERT | N$\times$|5.17 | 16.44 | 5.41 | 17.41 |
> | K = 10 (Stochastic)| 10$\times$ |6.44 | 17.71 | 6.61 | 18.32 |
> | K = 50% (Stochastic) | $\frac{N}{2} \times$ | 5.95 |17.04 |6.10| 17.58|
> | Transcormer| 3 $\times$ | 5.09 |16.30| 5.29 | 17.31 |
>
> We can find that our model significantly outperforms stochastic estimation. Besides, we also add the corresponding descriptions about stochastic estimation and distillation in the related works in the revision version to make a clear explanation.
>
> [a] Masked Language Model Scoring
>
> [b] Bert has a mouth, and it must speak: bert as a markov
>
> [c] Pre-Training Transformers as Energy-Based Cloze Models

---

> > ### Comment · Reviewer_ayVD · 2022-08-09
> > **Reply by Reviewer ayVD to Author Response**
> >
> > Thanks to the author for the improvements and further evaluations.
> >
> > **I raise my score from 4/10 to 6/10.** The work is thorough and introduces an architectural innovation to more richly model $P(y_s|y_{\backslash s})$ in one pass. Unlike the paper originally claimed, it is not the first such model (T-TA and Electric); also, unsupervised sentence scoring remains niche (MLM scoring is more for inspection of pre-existing models, and for downstream tasks most works use discriminative ranking). However, the authors have done many experiments, it performs better than MLMs and previous models, and may have applications elsewhere (analogous to XLNet, also published at NeurIPS). There are still many grammar errors; please work with a copyeditor or native English speaker on any camera-ready.
> >
> > Re: **Q1**: thanks for including T-TA, and for showing your gains over it (due to greater contextualization) in Table 8. You should make clearer that this is not the first model to do one-inference pass scoring (L53-L54), but it is the best (as shown by results) as it is the first with rich cross-token interaction. A comparison with T-TA on a second dataset should close remaining doubt.
> >
> > Re: **Q2, Q3, Q4, Q5, Q6,** thanks for the textual changes. Thanks for the commitment in **Q9**. Some final points:
> >
> > - You should still make clear that SLM has the same objective as MLM. You are changing the _architecture_, not the unsupervised metric. e.g., L31: "representative LMs" --> "representative LM training objectives"
> >
> > - L20-21 is better but still vague ("goodness"). You could say relative likeliness of a sentence, and then bring up where it _naturally_ arises: $P(y)$ in the Bayes rule decompositions of sequence-to-sequence tasks, i.e., $P(y|x) \propto P(x|y) P(y)$ (see Jelinek 1975, Brown 1993), and that higher $P(y)$ indicates more linguistically acceptable (i.e., per human judgment) sentences (Lau et al, 2017: "Grammaticality, acceptability, and probability: A probabilistic view of linguistic knowledge"). Then you can talk about more _heuristic_ uses (reranking in MT, ASR)
> >
> > - The mention of discriminative language modeling (L99-L100) should emphasize why it's different (requires a target downstream task)
> >
> > Thanks for the additional evaluations in **Q7, Q11, Q12** which affirm both the practicality and robustness of this method over existing ones. The discussions re: **Q8, Q10, Q12** also help clarify your thinking; make sure to refer readers to the Appendix from the main text.

---

> ### Author Response · Authors · 2022-08-02
> **To Reviewer ayVD**
>
> **Q8: Do you have an application where token-wise probability is essential.**
>
> Yes. There remain many potential applications that are heavily relied on token-wise probability. For example:
>
> 1. Correction (e.g., Chinese Spelling Error Correction [f], Grammatical error correction [d], ASR error correction [e]) task is to detect the error tokens and then correct these tokens, which means it needs to use the probability of each token to judge whether this token can match the remaining sentence and correct it (generate a new token) if it is incorrect. We leave it (applying SLM on correction) as future work. In other words, we think our SLM is quite suitable for solving post-processing tasks (e.g., reranking or correction).
>
> 2. Besides, we also want to highlight one of the core ideas of our SLM is to support the full prediction of the input sequence and both use the information of the whole sequence, which means it can be extended to pixel/patch-wise in image or frame-wise in speech, and then can be used for anomaly detection, which heavily depends on token-wise probability.
>
> 3. And in addition, we can convert NLU tasks as ranking classification. For example, in sentiment classification, we can formulate the input sentences as "the sentiment of <review> is positive" and "the sentiment of <review> is negative", and then calculate the sentence score. Generally, the sentence with targeted sentiment should have a lower log-probability score. We have conducted some discussions about this thinking in Appendix A.7.
>
> **Q9: Will you release pre-trained SLMs?**
>
> Of course. The official code/model release of Transcormer is on progress (including large pre-trained SLM), and besides we are also preparing a huggingface version for the community.
>
> **Q10: Are there any experiments to explain (SLM for scoring, CLM for NLG and MLM for NLU)?**
>
> Thanks for your question. In fact, although the query stream of SLM is similar as MLM which can also obtain bidirectional context, there still remain some differences in content between SLM and MLM. The query and content stream in MLM are shared together to learn bidirectional context, while SLM maintains three different streams and the content is composed of forward and backward streams. Therefore, the content in MLM is able to build bidirectional context while the content in SLM is used to collect forward and backward respectively. Considering that existing works [g] proved that BERT-style models can achieve state-of-the-art results in a fine-tuning fashion with bidirectional context, we think SLM is suitable for scoring and MLM prefers NLU tasks. But we also discuss the possibility of using SLM to solve NLU tasks in a prompt pattern and the discussions can refer to Appendix A.7.
>
> **Q11: Have you considered ways of matching MLM and SLM costs that avoid MLM pretraining mismatch?**
>
> Thank you for your question. Actually, we have tried to use smaller BERT or allow BERT to predict more or fewer tokens at the pre-training. But all of these attempts still cannot avoid the inherent issue in N-passes, which means the MLM-based model must need to forward N times to generate the probability of each token to obtain pseudo log-likelihood scores. We have conducted experiments about latency in Appendix. A.5. Just as shown in Table. 9, even using a smaller BERT model, it still requires 3.658s/1640s on GPU/CPU which is nearly 14.8x/82x computations over Transcormer (base) and will also affect model performance.  In other words, how to support efficient inference is the biggest challenge to limit the usage of the MLM-based model in sentence scoring.
>
> [d] Grammatical error correction using neural machine translation
>
> [e] Joint contextual modeling for asr correction and language understanding
>
> [f] Spelling error correction with soft-masked BERT
>
> [g] BERT: Pre-training of Deep Bidirectional Transformers for Language Understanding

---

> ### Author Response · Authors · 2022-08-02
> **To Reviewer ayVD**
>
> **Q12: More uses can be explored to justify this use.**
>
> Thank you for your suggestion. Our method has many potential applications besides reranking tasks:
>
> - First, to resolve your concerns, we have added the experiments on BLiMP to align [18], and the results are reported in Appendix A.3. And we find that our SLM can easily beat GPT model, and can outperform BERT performance slightly under the same parameters but just need single forward pass.
>
> - Just as mentioned in Q9, our SLM can is very suitable for post-processing tasks. For example, like correction or reranking tasks. Besides, our SLM can also be used for other modalities to produce pixel-wise or frame-wise probability. Specifically, some anomaly detection also heavily relied on token/pixel/frame-wise probability.
>
> - In addition, we can convert NLU tasks as a ranking classification by designing some prompt patterns, just as shown in Appendix A.7. Benefits from such design, we can easily conduct zero-shot inference at NLU tasks without any fine-tuning (Just like GPT-3). And we will continue to explore more techniques in this research direction.
>
> Besides, we note that both our SLM and MLM adopt masked tokens plus their positions to predict their targets, the main difference in context is that MLM adopts one bidirectional context and SLM adopts forward and backward contexts. We think that the MLM-based model with bidirectional context should also be able to produce good unidirectional context, so we can use the MLM-based model (e.g., BERT) for initialization to accelerate SLM training. In other words, we can also use these existing models to directly train our SLM, rather than from scratch. We find that our SLM, which is initialized with the BERT model, just needs 20K steps to match previous performance (i.e., training from scratch with 125k steps). The corresponding experiments are reported in Appendix A.4.

---

### Official Review · Reviewer_2vXa · 2022-07-15

**Rating:** 6
**Confidence:** 4
**Soundness:** 3 good
**Presentation:** 3 good
**Contribution:** 2 fair

**Summary:**

This work presents a Transformer-based model for sentence scoring, which aims to overcome the limitations of GPT-style causal language model pretraining (only the unidirectional context is used) and BERT-style masked language model pretraining (multiple forward passes are required when scores for each token are needed). The authors propose an XLNet-style "multi-stream" self-attention, which employs a forward stream (to capture the LTR context), a backward stream (for the RTL context), and a query stream which captures information from both.

This method is experimentally validated on two tasks, neural machine translation and reranking for ASR. The proposed method generally outperforms CLM and MLM approaches. The authors also perform further experiments showing that the proposed method is still able to outperform CLM when the size of the models is adjusted such that the amount of inference-time computation is equal; finally, they also attempt to make MLM more efficient by masking multiple tokens at a time (thereby reducing the amount of required computation), but the proposed method still comes out ahead.

**Questions:**

Could you clarify what the "oracle" line in Tabels 2 and 3 represents exactly?

**Limitations:**

I do not see any potential negative societal impacts of this work. Limitations are sufficiently addressed.

**Strengths And Weaknesses:**

The main idea seems like fairly straightforward variant of the XLNet architecture, not too dissimilar from other past proposals to modify its base architecture by changing the nature of the "streams".

Aside from originality, overall the paper feels solid. Both the architecture and the experiments are clearly described. The tasks chosen by the authors for the evaluation are realistic, and at least for the NMT experiments the baselines are sensible and their performance is in the right ballpark.

Given that one of the selling points of the proposed method is computational efficiency, the experiments of sections 5.1 and 5.2 are useful. The results are promising, showing that under comparable amounts of computation the authors' proposal still comes ahead.

A couple of minor points:
* It would be useful to have dataset and split sizes for the experiments of section 4 directly in the paper.
* IWSLT14 might be a small dataset, but I don't think German, Spanish, Italian, etc. should be described as "low-resource".

---

> ### Author Response · Authors · 2022-08-02
> **To Reviewer 2vXa**
>
> Thanks for your valuable comments. Below are our responses to your concerns:
>
> **Q1: It would be useful to have datasets and split sizes for the experiments of section 4 directly in the paper.**
>
> Thanks for your suggestions. The dataset and split sizes are described in Appendix A.1 and we have added the corresponding descriptions in Section 4.
>
> **Q2:  IWSLT14 might be a small dataset, but I don't think German, Spanish, Italian, etc. should be described as “low-resource”.**
>
> Thanks for your advice. We admit that it is not appropriate to describe these tasks as “low-resource”. We have adjusted the descriptions as “small-scale” in the revised version.
>
> **Q3: “Oracle” line in Table 2 and 3.**
>
> “Oracle” means calculating the performance between each candidate and ground truth, and then selecting the best one greedily as the oracle result. To some extent, “Oracle” means the upper bound of improvement space in the candidate set.

---

### Official Review · Reviewer_UGD7 · 2022-07-17

**Rating:** 3
**Confidence:** 3
**Soundness:** 2 fair
**Presentation:** 3 good
**Contribution:** 2 fair

**Summary:**

This paper proposes a transformer architectural variant. The motivation of this variant is that it contains a sliding language modeling objective for sentence scoring, that enables it to look at all the tokens in a sentence using bidirectional context in a single pass. This solves the issue of using standard causal language modeling or masked language models, which respectively have limitations of using only unidirectional context and requiring multiple forward passes.

**Questions:**

- The gains from SLM in the experiments required tuning a hyperparameter on a dev set and selecting the best value to use on the test set. Was the same hyperparameter value usable across all datasets? In what hyperparameter range was the model performing better than the baselines? What happens if we don't tune this hyperparameter?
- Why was NMT and ASR chosen instead of general language understanding, which how LMs such as BERT and GPT are mostly used today? If transcormer does better in those settings, then the impact of this work will be much greater. For instance, sentiment analysis can be formulated as scoring two sentences: "the sentiment of <review> is positive" and "the sentiment of <review> is negative". This is called "rank classification", and this is how GPT-3 is evaluated. While it is not necessary to re-train a GPT-3 size model, even on the small scale some experiments on language understanding tasks like these could improve the experimental validation of the paper immensely.
- I might have missed it but will the code be made publicly available to reproduce the experiments? This might be important to gain adoption for this method.
- Since there is a high-cost of re-training from scratch, is it possible for existing language models to be adapted for this new objective? This might enable more compute-efficient experiments.

**Limitations:**

The authors were up-front about the additional computational cost of their method, though additional experiments would be needed for the empirical gains in compute-matched settings to be convincing for me.

The societal impact part seems fine.

**Strengths And Weaknesses:**

## Strengths
- The structure of the paper is well-motivated. It is clear what the limitations of causal LM and MLM are for sentence scoring, and the paper proposes an architecture that addresses those two issues.
- The writing of the paper is quite clear. There is a natural progression from motivation to method to results, and overall the paper is pretty understandable.
- The significance of the paper is very high *in principle*. As causal LM and MLM are the standard objectives that are very commonly used in NLP, a new paradigm that outperforms these two types of objectives would be very impactful.
- The fact that SLM needs 3 times the compute of similar sized models is a limitation, but the authors explicitly explored this Section 5.

## Weaknesses
The promise of a better sentence scorer is potentially very impactful. The authors chose to evaluate sentence re-ranking on multiple generated candidates for NMT and for ASR. I have several concerns about these experiments that made them not particularly convincing for me.
- First, the experiments were, in my view, are a bit thin. For instance, the gain from SLM over BERT is not very large, and there is still a large gap between the oracle. Furthermore, while several baselines were provided, there have been many architectures recently proposed, and probably more comparisons are needed to fully justify the method. As some examples, it could have been good to compare with T5, BART, XLNet, etc.
- I am also not sure about the setup for the ablations. Why were the ablations not performed on all the datasets? I think this could have been addressed by putting a final row in Table 2, with SLM with one-third of the compute so that the compute is matched. It is fine if it does not win in all settings under the same compute, but in this case it would be good to be transparent about how much compute is needed to get the same performance, and whether there are performance gains in multiple different compute-matched settings.
- Finally, I am not sure why NMT and ASR were chosen as the experimental settings in the first place. It could have been stronger if the paper had chosen a language understanding benchmark like GLUE, which could be a better fit for pretrained language models.

I have voted for rejection primarily because the experiments are not particularly convincing, but if enough of the questions/concerns that I had are addressed in the rebuttal I will be happy to change my score.

---

> ### Author Response · Authors · 2022-08-02
> **To Reviewer UGD7**
>
> We sincerely thank the reviewer for providing valuable comments, which are helpful for improving our paper. Below are our responses to your concerns:
>
> **Q1: The gain from SLM over BERT is a bit thin.**
>
> Thank you for your question. Actually, the biggest advantage of our SLM over BERT is the efficiency (SLM just needs a single pass while BERT needs N-passes), as both SLM and MLM can utilize bidirectional context to predict token probability. In other words, SLM can achieve a much faster speed than BERT but also outperform BERT slightly in rescoring. To highlight the advantages of our SLM in efficiency, we conduct a series of experiments to investigate the inference latency between SLM and MLM at different sequence lengths. The results are shown in Appendix A.5 in the revision version and as follow:
>
> | Model | Params | GPU (sent = 500) | CPU (sent=500) |
> | :----- | :---- | ----: |----: |
> |BERT (small) | 34M  | 3658ms | 1650s |
> |BERT (base) | 110M | 7890ms | 3210s |
> |BERT (large) | 340M | 19770ms | 7433s |
> |Transcormer (small) | 34M | 183ms | 9.9s |
> |Transcormer (base) | 110M | 246ms | 20s |
>
> We can find that when the length of the sentence becomes very large (e.g., 500), MLM needs 7.9s/3210s at GPU/CPU in a base setting, which costs nearly 20x and 166x computations over SLM. These results validate the efficiency of our SLM when compared with MLM. Besides, we also add experimental results on BLiMP to further demonstrate the effectiveness of our SLM in calculating sentence scores.
>
> **Q2: Comparisons with some other baselines like T5, BART, XLNet and etc.**
>
> Thanks for your question. In the revision version, we have added the results of RoBERTa on BLiMP and ASR tasks for reference. Besides, for the sake of fairness, we do not compare with some strong pre-trained models (e.g., RoBERTa, XLNet, T5, BART) directly, since these models cost more computations than the standard BERT model (e.g., RoBERTa trains on 160GB corpus with 500K steps at a batch size of 8192 while BERT is trained on 16GB corpus with 1M steps at a batch size of 256). Therefore, these models can easily produce better representations, but we believe that our SLM can also outperform their performance under the same training hyper-parameters, which we leave for the future work.
>
> **Q3: Ablation study on all the datasets**
>
> Thanks for your suggestion. We have added the ablation study on all the datasets in the revised version.
>
> **Q4: why not try some language understanding tasks?**
>
> Thank you very much for your comments.
>
> Originally, due to the mismatch between MLM and SLM, we do not try language understanding tasks directly since the query and content streams in the MLM-based model are shared together to enjoy the benefits of the bidirectional context, while SLM maintains query and forward/backward streams individually. Existing works [a][b][c][d] have proven that BERT-style models can achieve advanced results in NLU tasks in a fine-tuning fashion due to the benefits of bidirectional context. Although the query stream of SLM is able to enjoy completely bidirectional context like MLM, the content stream of SLM can only learn unidirectional context, which may be weaker than the MLM-based model in NLU tasks if using bidirectional context for fine-tuning.
>
> We have tried to fine-tune our Transcormer like BERT model (i.e., fine-tune in a bidirectional context) in our internal experiments on SST-2. Our model only produces 91.9\% while BERT can achieve 92.8\%. We argue this is because of the mismatch (forward/backward context v.s. bidirectional context) between the pre-training and fine-tuning.
>
> However, thank you very much for reminding us that we can convert classification tasks into rank classification via designing prompt-based templates. We have tried some different prompts in our experiments and the results are shown in Appendix A.7. We find that designing a prompt like `` [Review] is a [label] sentiment.” can produce 71.0% accuracy on SST-2 in a zero-shot setting (without fine-tuning). We think it is very interesting and we will continue to explore existing techniques and design more advanced techniques based on such a paradigm to discover the potential applications of using SLM in solving NLU tasks. We will also report the latest results about this branch during the discussion stage if it is done. We will also acknowledge your advice in our final version.
>
> [a] BERT: Pre-training of Deep Bidirectional Transformers for Language Understanding
>
> [b] XLNet: Generalized Autoregressive Pretraining for Language Understanding
>
> [c] RoBERTa: A Robustly Optimized BERT Pretraining Approach
>
> [d] ALBERT: A Lite BERT for Self-supervised Learning of Language Representations

---

> > ### Comment · Reviewer_UGD7 · 2022-08-08
> > **Thanks for the revisions**
> >
> > > Actually, the biggest advantage of our SLM over BERT is the efficiency
> >
> > Thanks for this clarification
> >
> > > We believe that our SLM can also outperform [T5, BART, XLNet and etc]'s performance under the same training hyper-parameters, which we leave for the future work.
> >
> > Maybe? Seems pretty speculative... Could be good to say this specifically in the paper.
> >
> > > Language understanding tasks
> >
> > Given that the NLU tasks are still pretty weak, I am not super convinced about the experimental benefits of this method. It might benefit to clarify more what scenarios it will be useful it it cannot be used for something like sentiment analysis. Maybe it's good to continue SuperGLUE evals and show the results. Inference is fast using the author's proposed method, so it should be easy to do this.
> >
> > Overall, while I appreciate the additional experiments. I can't increase my score. I'd increase my score to 3.5 if I could (something like, I think the idea is nice but the experimental benefits are still quite weak, doesn't generalize to a lot of settings, only a small subset of people would be interested). However, this paper, even with the rebuttal, is not quite at the score of 4 for me. My opinion is that the paper would make for a much more impactful future submission with more evaluations, for example on SuperGLUE tasks, which would make this model of interest to a wider community. I hope the authors will be able to write an even better paper moving forward (if they wish, and if this paper is not accepted) that will later have much stronger impact due to more convincing evaluations

---

> ### Author Response · Authors · 2022-08-02
> **To Reviewer UGD7**
>
> **Q5: Was the same hyperparameter value usable across all datasets? what hyperparameter range was the model performing better than the baselines? What happens if we don't tune this hyperparameter?**
>
> Thanks for your question. Generally, for seq2seq (e.g., NMT and ASR) tasks, we need to consider the effect of the original score produced by seq2seq models and language model score on the generated data. In other words, it is necessary to tune a hyper-parameter to balance the effect of the seq2seq model and the language model, and this strategy has been widely used in many works [e][f][g][h]. Generally, the value of hyper-parameter is sensitive to the performance of the original score, and in my experiments, for NMT tasks, the hyper-parameter $\lambda$ is between 0.1 ~ 0.2 and for ASR tasks, the hyper-parameter $\lambda$ is between 0.5 ~ 3.0. Directly scoring sentences without hyper-parameter cannot improve performance for seq2seq tasks. But for single-sentence tasks (e.g., BLiMP), since it only needs to evaluate the quality of input sequences, we do not need hyper-parameter anymore and the experimental results on BLiMP have been reported in Appendix. A.3.
>
>
> **Q6: Code release?**
>
> Of course. The code has been attached to the supplementary material for reference. The official code/model release of Transcormer/SLM is in progress (including pre-trained large SLM), and besides we are also preparing a huggingface version for the community.
>
> **Q7: Is it possible for existing language models to be adapted for this new objective?**
>
> Yes, we think there are some different strategies for existing language models to be adapted for our SLM:
>
> - First, we can directly use the MLM-based/Masked-based model for initialization and continue to train our SLM for acceleration. Since we note that both our SLM and MLM adopt masked tokens plus their positions to predict their targets, the main difference in context is that MLM adopts one bidirectional context and SLM adopts forward and backward contexts. We think that the MLM-based model with bidirectional context should also be able to produce good unidirectional context and SLM does not modify the model structure, so we can use the MLM-based model (e.g., BERT) for initialization to accelerate SLM training. In other words, we can also use these existing models to directly train our SLM, rather than from scratch. We find that our SLM, which is initialized with the BERT model, just needs 20K steps to match previous performance (i.e., training from scratch with 125k steps). The corresponding experiments are reported in Appendix. A.4.
>
> - For T5, BART or other encoder-decoder frameworks, we can directly replace the objective of the decoder as our SLM objective. For example, we can mask tokens or segments in the sequence used in the encoder and for the masked parts, we can use SLM to calculate each token in the decoder. This approach can be viewed as a seq2seq reranker.
> Besides, we also believe there also remain some other techniques to adopt existing language models for our SLM and we will continue to explore the possibility of this branch.
>
> Besides, we also believe there also remain some other techniques to adopt existing language models for our SLM and we will continue to explore the possibility of this branch.
>
> [e] Masked Language Model Scoring
>
> [f] Facebook FAIR's WMT19 News Translation Task Submission
>
> [g] Effective Sentence Scoring Method Using BERT for Speech Recognition
>
> [h] Fast and Accurate Deep Bidirectional Language Representations for Unsupervised Learning

---

### Author Response · Authors · 2022-08-02
**To All Reviewers**

We sincerely thank each reviewer for providing constructive comments for our paper, which are very helpful to improve our paper. Below are our modifications to the latest version:
1. Following the suggestions of Reviewer UGD7, we have added the ablation studies of all datasets in the revised version and removed the original Table 4.
2. Following the suggestions of Reviewer ayVD, we have modified the descriptions of “quality of a sentence” and “precise sentence scoring” in the introduction. We also move the definitions of CLM and MLM to the introduction.
3. Following the suggestions of Reviewer ayVD, we have added more related works (e.g., T-TA and Electric) in the background and moved the descriptions about discriminative rescoring in the main text. Due to the page limitations, we do not present many introductions about each relate work in the main text and put them in Appendix A.2.
4. We have added Appendix A.2 to introduce more details about related works.
5. We have added Appendix A.3 to report results on BLiMP dataset and the comparison with other works.
6. We have added Appendix A.4 to introduce how to accelerate the training of SLM.
7. We have added Appendix A.5 to describe the inference latency between SLM and MLM.
8. We have added Appendix A.7 to discuss the potential of using SLM in language understanding tasks.
9. We also fix the minor notes and writing problems pointed out by each reviewer.

---

### Author Response · Authors · 2022-08-08
**To All Reviewers**

Dear Reviewers, thank everyone to provide your valuable comments. Since the deadline of the Author- Reviewer Discussion is nearly to end, we are looking forward to the feedback of all of you.

---

### Meta-Review · Area_Chair_6VVH · 2022-08-29

**Recommendation:** Accept
**Confidence:** Less certain

**Metareview:**

This paper proposes a transformer architectural variant. The motivation of this variant is that it contains a sliding language modeling objective for sentence scoring, that enables it to look at all the tokens in a sentence using bidirectional context in a single pass. This solves the issue of using standard causal language modeling or masked language models, which respectively have limitations of using only unidirectional context and requiring multiple forward passes.
The empirical gains in terms of quality are modest, but the speed-ups are quite impressive. It would have been nice to see evaluations on a few more tasks (like SuperGLUE). It is a bit unclear why such results are not presented, but on balance the paper is probably still above the cut-off.

**Award:**

No

---

### Decision · Program_Chairs · 2022-09-14

Accept